# Undersampled Diffusion-Weighted ^129^Xe MRI Morphometry of Airspace Enlargement: Feasibility in Chronic Obstructive Pulmonary Disease

**DOI:** 10.3390/diagnostics13081477

**Published:** 2023-04-19

**Authors:** Samuel Perron, David G. McCormack, Grace Parraga, Alexei Ouriadov

**Affiliations:** 1Department of Physics and Astronomy, The University of Western Ontario, London, ON N6A 3K7, Canada; 2Division of Respirology, Department of Medicine, The University of Western Ontario, London, ON N6A 3K7, Canada; 3Robarts Research Institute, London, ON N6A 5B7, Canada; 4Department of Medical Biophysics, The University of Western Ontario, London, ON N6A 3K7, Canada; 5Graduate Program in Biomedical Engineering, The University of Western Ontario, London, ON N6A 3K7, Canada

**Keywords:** acceleration, lung, morphometry, hyperpolarized, Xenon, compressed sensing, alpha-1 antitrypsin deficiency, COPD

## Abstract

Multi-b diffusion-weighted hyperpolarized gas MRI measures pulmonary airspace enlargement using apparent diffusion coefficients (ADC) and mean linear intercepts (*L*_m_). Rapid single-breath acquisitions may facilitate clinical translation, and, hence, we aimed to develop single-breath three-dimensional multi-b diffusion-weighted ^129^Xe MRI using k-space undersampling. We evaluated multi-b (0, 12, 20, 30 s/cm^2^) diffusion-weighted ^129^Xe ADC/morphometry estimates using a fully sampled and retrospectively undersampled k-space with two acceleration-factors (AF = 2 and 3) in never-smokers and ex-smokers with chronic obstructive pulmonary disease (COPD) or alpha-one anti-trypsin deficiency (AATD). For the three sampling cases, mean ADC/*L*_m_ values were not significantly different (all *p* > 0.5); ADC/*L*_m_ values were significantly different for the COPD subgroup (0.08 cm^2^s^−1^/580 µm, AF = 3; all *p* < 0.001) as compared to never-smokers (0.05 cm^2^s^−1^/300 µm, AF = 3). For never-smokers, mean differences of 7%/7% and 10%/7% were observed between fully sampled and retrospectively undersampled (AF = 2/AF = 3) ADC and *L*_m_ values, respectively. For the COPD subgroup, mean differences of 3%/4% and 11%/10% were observed between fully sampled and retrospectively undersampled (AF = 2/AF = 3) ADC and *L*_m_, respectively. There was no relationship between acceleration factor with ADC or *L*_m_ (*p* = 0.9); voxel-wise ADC/*L*_m_ measured using AF = 2 and AF = 3 were significantly and strongly related to fully-sampled values (all *p* < 0.0001). Multi-b diffusion-weighted ^129^Xe MRI is feasible using two different acceleration methods to measure pulmonary airspace enlargement using *L*_m_ and ADC in COPD participants and never-smokers.

## 1. Introduction

Hyperpolarized ^129^Xe pulmonary MRI [1,2] provides physiologically relevant biomarkers of obstructive lung disease [3,4,5]. Recently, the ^129^Xe MRI pulmonary apparent diffusion coefficient (ADC) [6] was shown to be strongly related to ex vivo histological measurements of airspace enlargement in lung tissues harvested from COPD participants. In addition, ^129^Xe ventilation MRI was shown to be feasible using naturally abundant ^129^Xe [7]. Moreover, dissolved-phase ^129^Xe MRI may be employed for simultaneous ventilation/perfusion lung imaging [8,9,10,11], and there is a stable supply of ^129^Xe and commercially available polarizers capable of generating the necessary volumes of highly polarized gas for clinical investigations [12,13]. However, the low gyromagnetic ratio of ^129^Xe and the gradient strengths typical for clinical scanners (5 G/cm) dictate that rapid MRI acquisition strategies be developed [9]. This is especially true for multi-b diffusion-weighted MRI, as whole lung datasets are currently difficult to acquire during the relatively short 10–18 s breath hold timeframe [14] that is feasible in participants with lung disease. For participants with obstructive lung disease stemming from abnormal lung airspace enlargement, multi-b diffusion-weighted MRI [15,16] provides ADC and other acinar duct morphometric measurements that estimate mean linear intercept (*L*_m_) values. This is important for young adults with bronchopulmonary dysplasia (BPD) [17] and alpha-1 antitrypsin deficiency (AATD) [18] in whom preliminary diffusion-weighted MRI studies [19,20] have been performed. Because emphysema leads to the destruction of the lung microstructure and a subsequent decrease in surface area of the alveolar walls, gas motion in the lungs is less restricted, leading to higher ADC values. Moreover, since *L*_m_ is inversely proportional to the lung surface-to-volume ratio, destruction or damage of the lung microstructure leads to larger *L*_m_ values. As ADC and *L*_m_ values increase, normal/healthy gas exchange diminishes, making these measurements clinically practical for evaluating lung function in emphysema.

Three-dimensional multi-b diffusion-weighted MRI [21], requiring a number of independent doses of gas for each slice or each b value, has been performed using ^129^Xe [14] and ^3^He [21,22]. While all these approaches are feasible in the research setting, the increased time for acquisition, the potential for lung volume mismatch, and repeated doses of hyperpolarized gas are not compatible with clinical examinations. Half-Fourier RARE-type or TrueFISP [23,24], parallel imaging [25], simultaneous slice acquisition [26], and compressed sensing (CS) [27] are promising options for decreasing image acquisition time. Recently, ^3^He ventilation MRI was shown using CS [27], and multi-b diffusion-weighted MRI was demonstrated using conventional k-space sampling [28], parallel imaging [29], CS [30] and undersampling in the spatial and in *b*-value directions [31].

We hypothesized that, by using k-space undersampling, whole lung three-dimensional multi-b diffusion-weighted ^129^Xe MRI can be achieved in a single 16 s breath-hold. Therefore, in this proof-of-concept evaluation, our objective was to retrospectively evaluate and compare ADC and morphometry estimates [32,33] in never-smokers and COPD study participants using partial Fourier reconstruction [34] and compressed sensing [35].

## 2. Theory

The signal dependence related to diffusion-sensitization can be determined through the probability density function or diffusion propagator (*P*) for fluid diffusion in confined media with unknown geometry [20,32,33]:(1)$$S\left(b\right)/S_{0}=\overset{\infty }{\underset{0}{\int }}P\left(D\right)\exp\left(-D\cdot \bar{b}\right)dD$$
where *D* is diffusivity, *S*(*b*) is the signal at a particular *b*-value, and *S*_0_ is the MR signal-intensity in the absence of diffusion-sensitizing gradients. The diffusion propagator can be ascertained through the inverse Laplace transform of *S*(*b*) [33]. To apply this, the analytical representation for *S*(*b*) is required. Thus, experimental *S*(*b*) values can be fitted, as demonstrated for multi-b diffusion-weighted ^3^He MRI [20,30,31] as follows:(2)$$S\left(b\right)/S_{0}=\exp{\left(-D^{\prime }\cdot \bar{b}\right)}^{\alpha }$$
where *D*′ is the apparent diffusivity and *α* is the heterogeneity index (0 < *α* ≤ 1.0). The diffusion propagator can be determined through substitution of Equation (2) into Equation (1) and then applying the inverse Laplace transform [33]:(3)$$P\left(D\right)=\frac{B/D’}{{\left(D/D’\right)}^{\left(1-\alpha /2\right)/\left(1-\alpha \right)}}\exp\left[-\frac{\left(1-\alpha \right)\alpha^{\frac{\alpha }{\left(1-\alpha \right)}}}{{\left(\frac{D}{D^{\prime }}\right)}^{\frac{\alpha }{\left(1-\alpha \right)}}}\right]f\left(D\right)$$
(4)$$\mathrm{and}\;f\left(D\right)=\left\{\begin{array}{c} 1/\left[1+C{\left(D/D’\right)}^{\left(0.5\alpha -\alpha^{2}\right)/\left(1-\alpha \right)}\right],\;\;\alpha \leq 0.5\\ \;\;\;\;\;\;\;\;1+C{\left(D/D’\right)}^{\left(0.5\alpha -\alpha^{2}\right)/\left(1-\alpha \right)},\;\;\alpha >0.5 \end{array}\right.$$
where *f*(*D*) is the auxiliary function, and parameters *B* and *C* are functions of the heterogeneity index [33]. Mean *D* estimates can be determined using the probability density function distribution to calculate mean airway length maps [32] (*L*m_D_ = $\sqrt{2\Delta D}$, where Δ is the diffusion time). For multi-b diffusion-weighted ^3^He MRI, *L*_m_ is empirically observed to be proportional to *L*m_D_ [20]:(5)$$L_{m}=-562\;\mu m+4.3\cdot Lm_{D}$$

Mean airway length depends on both Δ and diffusivity, so Equation (5) cannot be used for ^129^Xe MRI-based *L*_m_ estimates. In order to extend Equation (5) to ^129^Xe gas, the empirical relationship in Equation (6) was previously determined and proposed [36]:(6)$$L_{m}=-562\;\mu m+4.3\cdot Lm_{D}\cdot \sqrt{\frac{{2D}_{0}^{\mathrm{He}}\Delta_{\mathrm{He}}}{{2D}_{0}^{\mathrm{Xe}}\Delta_{\mathrm{Xe}}}}$$
where $D_{0}^{\mathrm{He}}$ is the free diffusion coefficient of ^3^He in a nitrogen gas mixture (0.88 cm^2^/s), Δ_He_ = 1.46 ms [20], $D_{0}^{\mathrm{Xe}}$ is the free diffusion coefficient of ^129^Xe (0.12 cm^2^s^−1^/0.14 cm^2^s^−1^ [37]), and Δ_Xe_ is the diffusion time. To validate our approach, a single participant (COPD-5) was evaluated using both ^3^He and ^129^Xe MRI. Stretched-exponential model ^3^He MRI *L*_m_ values for participant COPD-5 (*L*m_D_ = 290 ± 50 µm and *L*_m_ = 700 ± 180 µm) were similar to previously published estimates [20].

## 3. Materials and Methods

### 3.1. Study Participants

Four never-smokers, four COPD ex-smokers with emphysema, and one AATD study participant with COPD provided written informed consent to an ethics board-approved protocol (The University of Western Ontario Health Sciences Research Ethics Board, approval ID 18130 and 18131) that was compliant with the Health Insurance Portability and Accountability Act (HIPAA, USA). Ex-smokers with COPD and AATD participants with COPD were enrolled between 50–80 years of age as part of the Thoracic Imaging Network of Canada (TinCan) cohort [38]; never-smokers without a history of tobacco smoking or chronic respiratory disease were enrolled between 45–80 years of age. Some of the participants evaluated here were previously reported [14,37] as a part of the TINCan study.

### 3.2. Pulmonary Function Tests and CT

Spirometry, plethysmography, and the diffusing capacity of the lung for carbon-monoxide (DL_CO_) were performed according to American Thoracic Society (ATS) guidelines [39] using a plethysmograph and attached gas analyzer (MedGraphics Corporation. 350 Oak Grove Parkway, Saint Paul, MN, USA). CT was also performed in supine (64-slice Lightspeed VCT scanner GEHC, Milwaukee, WI, USA; 64 × 0.625 mm, 120 kVp, effective mA = 100, tube rotation time = 500 ms, pitch = 1.0), using a spiral acquisition in breath-hold after inhalation of 1 L N_2_ from functional residual capacity (FRC). A slice thickness of 1.25 mm and a standard convolution kernel were used.

### 3.3. ^129^Xe and ^3^He MRI Acquisition

MRI was performed at 3.0 T (MR750, GEHC, Waukesha, WI, USA) using whole-body gradients (*G_max_* = 5 G/cm, slew rate = 200 mTm^−1^s^−1^), as previously described [14]. ^129^Xe gas (86% enriched, measured polarization 12–40%) was provided by commercial polarizer systems (XeBox-E10, Xemed LLC, Durham, NH; XeniSpin™, Polarean Inc, Durham, NC, USA). All participants inhaled 1 L of a 50/50 by volume ^129^Xe/^4^He gas mixture from functional residual capacity (FRC). For all ^3^He MRI, polarized ^3^He gas (polarization ~40%) was provided by a commercial system (Helispin^TM^, Polarean Inc, Durham, NC [40]). Participants inhaled 1 L of a ^3^He/N_2_ a gas mixture (30/70 by volume) from FRC, as previously described [38].

A three-dimensional FGRE sequence was employed for ventilation MRI (15 mm coronal slices, matrix size 80 × 128, total acquisition time 16 s, 0.5 ms rectangular RF pulse, variable flip angle (VFA), TE/TR = 2.0 ms/9 ms, bandwidth = 24.5 kHz, FOV = 40 × 40 cm^2^), as previously described [3]. For ^3^He MRI, a multi-slice interleaved centric two-dimensional FGRE diffusion-weighted sequence was acquired for seven 30 mm coronal slices (matrix size = 128 × 80, total acquisition time = 16 s, 0.9 ms selective RF pulse, TE/TR = 3.9 ms/5.6 ms, bandwidth = 62.5 kHz, FOV = 40 × 40 cm^2^, *b* = 0, 1.6, 3.2, 4.8, 6.4 s/cm^2^). The diffusion-sensitization gradient pulse ramp up/down time = 0.5 ms, with Δ_He_ = 1.46 ms, which was initiated at the maximum *b* value to ensure that maximum MR signal was acquired at greater *b* values, as previously described [20].

For ^129^Xe MRI in the AATD participant, four interleaved acquisitions (3D FGRE, 0.5 ms rectangular RF pulse, VFA, TE/TR = 9.0 ms/10.0 ms, matrix size = 64 × 64, number of slices = 7; slice thickness = 30 mm, and FOV = 40 × 40 cm^2^), with and without diffusion-sensitization, were acquired for a given line of k-space. For participants with COPD and never-smokers, two interleaved acquisitions (2D FGRE, TE/TR = 9.8 ms/11.0 ms, matrix size = 128 × 128, number of slices = 7; slice thickness = 30 mm, and FOV = 40 × 40 cm^2^), with and without diffusion–sensitization, were acquired for a given line of k-space to ensure that RF depolarization (5° constant flip angle was used), and *T*_1_ relaxation effects were minimal [14]. In all cases, the diffusion–sensitization gradient pulse ramp up/down time = 500 μs, constant time = 2 ms, and Δ_Xe_ = 5 ms, providing four *b* values 0, 12.0, 20.0, and 30.0 s/cm^2^. ^3^He/^129^Xe MRI scans for the AATD participants were conducted back-to-back.

### 3.4. Image Analysis

As shown in Figure 1, two k-space masks, mimicking the acceleration factor 2 (AF = 2; half-echo sampling in the phase-encoding direction), and AF = 3, were applied to the fully sampled multi-b diffusion-weighted ^129^Xe k-space data. A partial Fourier reconstruction (MATLAB R2013b MathWorks, Natick, MA, USA) [41] was used to reconstruct diffusion-weighted images with AF = 2, while compressed sensing (MATLAB) [35] was used to reconstruct diffusion-weighted images with AF = 3. All fully sampled k-space data for ^129^Xe MRI were reconstructed into a 128 × 128 matrix (Fourier transform, IDL6.4, ITT Visual Information Solutions, Boulder, CO, USA). Therefore, the nominal voxel size was 3.1 × 3.1 × 15 mm^3^ for the static ventilation images and 3.1 × 3.1 × 30 mm^3^ for the diffusion-weighted images. Diffusion-weighted ^3^He MRI k-space data were zero-filled to a 128 × 128 matrix and then Fourier transformed (IDL6.4). Therefore, the nominal image voxel size was 3.1 × 3.1 × 30 mm^3^.

To generate *D*′ and *α* maps, a nonlinear least squares algorithm (MATLAB) was used to fit Equation (2) on a voxel-by-voxel basis. In turn, *D*′ and *α* were used to compute *P*(*D*) for the 0 < *D* ≤ *D*_0_ interval based on Equations (3) and (4) on a voxel-by-voxel basis (IDL 6.4).

A semi-automated segmentation approach was used to generate ventilation defect percent (VDP), as previously described [42]. ADC maps were generated for two *b*-values (0 and 12 s/cm^2^) on a voxel-by-voxel basis, as previously described [3] The relative area of the CT density histogram with attenuation values ≤−950 Hounsfield units (RA_950_) [43] and low attenuating clusters were determined using Pulmonary Workstation 2.0 (VIDA Diagnostics Inc., Coralville, IA, USA).

### 3.5. Statistics

Differences between ADC and *L*_m_ values generated from fully and undersampled k-space were calculated on a voxel-by-voxel basis [30] using Equation (7):(7)$$\mathrm{Difference}=\overset{N}{\underset{i=1}{\sum }}\overset{M}{\underset{j=1}{\sum }}\left[\frac{{\mathrm{FullySampled}}_{\mathrm{ij}}{-\mathrm{UnderSampled}}_{\mathrm{ij}}}{{\mathrm{FullySampled}}_{\mathrm{ij}}}\right]\times 100\%$$
where N and M were the corresponding map matrix sizes. Multivariate analysis of variance (MANOVA) and independent *t* tests were performed using SPSS Statistics, V22.0 (SPSS Inc., Chicago, IL, USA). For all participants and for the COPD subgroup, repeated measures of ANOVA with AF = [1, 2, and 3] as repeated ADC/*L*_m_ measurements were corrected using a Greenhouse-Geisser correction and used to determine any main effects for the acceleration factor regarding ADC or *L*_m_. Relationships between voxel-wise ADC and *L*_m_ with acceleration factor were determined using Spearman correlation coefficients (ρ). Agreement between acceleration factors for both ADC and *L*_m_ were determined using the Bland-Altman method [44] by using GraphPad Prism version 7.00 (GraphPad Software Inc., San Diego, CA, USA). Results were considered significant when the probability of two-tailed type I error (α) was less than 5% (*p* < 0.05).

## 4. Results

Table 1 summarizes pulmonary function, CT, and demographic measurements for all participants. Never-smokers reported significantly different FEV_1_/FVC, TLC, and DL_CO_ as compared to COPD participants.

Figure 2 shows the representative centre coronal ^129^Xe ADC and *L*_m_ maps for a single never-smoker, as well as COPD and AATD participants using all three approaches (fully sampled, AF = 2, 3). As shown in Figure 2, in the elderly never-smoker (FEV_1_ = 105%_pred_, DL_CO_ = 94%_pred_, RA_950_ = 0.14%, VDP = 4%), ADC and diffusivity maps were homogeneous, and the *L*_m_ maps were contiguous, and undersampled (AF = 2 and 3) maps were qualitatively similar to the fully sampled map. For the COPD participant (FEV_1_ = 59%_pred_, DL_CO_ = 43%_pred_, RA_950_ = 12% and VDP = 15%) with severe emphysema, ADC and *L*_m_ maps were qualitatively similar in relation to the undersampled and fully sampled cases. For the representative AATD participant with severe emphysema (FEV_1_ = 58%_pred_, DL_CO_ = 50%_pred_, RA_950_ = 19%, VDP = 27%), all k-space sampling methods provided continuous and qualitatively similar lung ADC and *L*_m_ maps.

Table 2 and Appendix A summarizes mean ADC, *L*m_D_, and *L*_m_, as well as *D*′ and *α*, for all participants; a by-participant list of these data is provided in a Appendix A (online). For all three k-space sampling methods, ADC and *L*_m_ were significantly different in the COPD subgroup 0.08 cm^2^s^−1^/580 µm, AF = 3; all *p* < 0.05) as compared to never-smokers (0.05 cm^2^s^−1^/300 µm, AF = 3), concomitant with abnormal airspace enlargement in COPD participants. For never-smokers, mean differences of 7%/7% for ADC and 10%/7% (AF = 2/AF = 3) for *L*_m_ values were observed compared to fully sampled values. For the COPD subgroup, mean differences (AF = 2/AF = 3) of 3%/4% for ADC and 11%/10% for *L*_m_ were observed between fully sampled and undersampled values. For the AATD participant, there was a mean difference of 7%/8% and 13%/12% (AF = 2/AF = 3) for ADC and *L*_m_ values, respectively.

For all participants (*p* = 0.9) and for the COPD subgroup (*p* = 0.4), ANOVA showed that there was no relationship between acceleration factor and ADC or *L*_m_. Figure 3A,B shows that AF-1 ADC values were significantly correlated with AF-2 (r^2^ = 0.82, ρ = 0.90, *p* < 0.0001, y = 0.88x + 0.008) and AF-3 (r^2^ = 0.86, ρ = 0.92, *p* < 0.0001, y = 0.87x + 0.008) values, and there was strong agreement for ADC AF-1 with AF-2 (bias = −0.0002 ± 0.01 cm^2^/s, lower limit = −0.02 cm^2^/s, upper limit = 0.02 cm^2^/s) and AF-3 (bias = −0.0007 ± 0.01 cm^2^/s, lower limit = −0.02 cm^2^/s, upper limit = 0.02 cm^2^/s). Figure 3C also shows ADC differences for AF-1 with AF-2 and AF-3 for each participant. Figure 3D,E shows that AF-1 *L*_m;_ was significantly correlated with AF-2 (r^2^ = 0.64, ρ = 0.75, *p* < 0.0001, y = 0.79x + 101) and AF-3 (r^2^ = 0.66, ρ = 0.76, *p* < 0.0001, y = 0.77x + 78) values, with strong agreement for *L*_m_ AF-1 with AF-2 (bias = 8 ± 146 μm, lower limit = −278 μm, upper limit = 293 μm) and AF-3 (bias = −24 ± 141 μm, lower limit = −300 μm, upper limit = 252 μm). Figure 3F shows *L*_m_ differences for AF-1 with AF-2 and AF-3 for each participant.

## 5. Discussion

Hyperpolarized gas ^129^Xe MRI was approved by the Food and Drug Administration (FDA) for clinical use in December 2022, opening the doors for wide clinical adoption and usage of this imaging modality. Diffusion-weighted hyperpolarized gas ^129^Xe MRI, along with static ventilation and gas exchange measurements, should be useful for the diagnosis, observation, and treatment outcome assessment of various pulmonary diseases, including smoking-related emphysema, AATD, and bronchopulmonary dysplasia. As ^129^Xe MRI is a radiation-free non-invasive imaging modality, it could potentially become the main lung imaging method for young adults and newborns.

In this proof-of-concept study, we investigated the influence of k-space undersampling on ^129^Xe MRI ADC and *L*_m_ values using three different sampling approaches and a stretched exponential model. We retrospectively evaluated nine participants, including never-smokers, COPD ex-smokers, and a single AATD participant to explore the feasibility of this approach.

COPD ex-smokers and AATD participants were previously studied using ^3^He MRI ADC and morphometry measurements [20,45,46,47,48]. However, to our knowledge, this is the first demonstration of undersampled ^129^Xe MRI ADC and *L*_m_ across a spectrum of emphysema severity and using the stretched exponential method. Previous ^129^Xe morphometry studies [49,50], in mainly healthy participants, provided a framework for this examination in participants with emphysema. Several factors are expected to influence ADC and *L*_m_ values, such as severity of emphysema and lung aging, due to the fact these measurements are an indirect reflection of the lungs’ and alveoli’s ability to move and transfer gasses: *L*_m_ is the lung microstructure dimension and ADC describes the motion of gasses within the lungs and the airway restrictions. As such, any destruction of the airways and alveoli leads to a change in ADC and *L*_m_ values, but the destruction pattern and the distribution of these ADC and *L*_m_ values differ from normal lung aging compared to emphysematous lungs [28].

Across a wide range of emphysema severity, partial Fourier reconstruction (AF = 2) and compressed sensing (AF = 3) did not significantly alter ADC and *L*_m_ values (*p* > 0.05) compared to those generated using fully-sampled Fourier transform reconstruction. In other words, for the never-smoker and COPD subgroups, fully and under-sampled estimates of ADC, *L*m_D_, and *L*_m_ were not significantly different. For both participant subgroups, the difference in *L*_m_ values calculated based on Equation (7) was the same for AF = 2/AF = 3, indicating that the two different image reconstruction methods led to similar morphometry estimates. Moreover, for all participants and for the COPD subgroup, there was no relationship for acceleration factor with ADC or *L*_m_. The strong and significant voxel-wise correlations for AF-1 ADC and *L*_m_ values with AF-2 and AF-3 values also support the notion that undersampling does not alter or bias ADC or *L*_m_ values and can be considered for participant studies.

It is important to note that, for the never-smoker subgroup, *L*m_D_ estimates were smaller (140 µm vs. 160 µm) than previous estimates [49] for four healthy volunteers at 1.5 T (same Δ = 5 ms and *b*-values). The difference may be due to the smaller $D_{0}^{\mathrm{Xe}}$ used here and previously described [37] (0.12 cm^2^/s vs. 0.14 cm^2^/s [49]), and this further demonstrates the need for a time- and gas-independent airspace morphometry estimate, such as *L*_m_. The correlation between DL_CO_ and ADC values has previously been studied by Kirby et al. [51], but such a correlation has not yet been reported for *L*_m_ values.

The empirical equation (Equation (6)) was developed based on Equation (5), which, in turn, was previously validated in COPD participants with a wide range of emphysema severity [20]. For the single AATD participant evaluated here using both ^3^He and ^129^Xe, mean ^3^He MRI *L*_m_ estimates [20], based on Equation (5), and mean ^129^Xe MRI *L*_m_ estimates, based on Equation (6), were similar (700 ± 180 µm vs. 690 ± 210 µm, for ^3^He and ^129^Xe, respectively); the approximate 1–2% difference in ^3^He- and ^129^Xe-derived *L*_m_ values likely reflected potential differences in slice location and in-plane resolution. At the same time, for the AATD participant, there was a 25% difference between ${\mathrm{Lm}}_{D}^{\mathrm{He}}$ and ${\mathrm{Lm}}_{D}^{\mathrm{Xe}}$ estimates (290 ± 50 µm vs. 220 ± 40 µm, for ^3^He and ^129^Xe, respectively), likely due to the Δ/D_0_ dependence of the mean airway length scale [49]. Unfortunately, there are no published ^129^Xe *L*_m_ values for comparison in participants with similar age or with the image matrix size used here. Previous ^129^Xe *L*_m_ estimates for young healthy volunteers were smaller [52] than the values reported here (200 µm vs. 280 µm), which is consistent with the fact that the participants evaluated here were significantly older than in the previous study. ^129^Xe *L*_m_ values for never-smokers and COPD participants were not significantly different from the SEM-based *L*_m_ values estimated using ^3^He MRI previously reported [20]; this suggests that Equation (6) may be considered for ^129^Xe MRI SEM morphometry estimates, although the relationship between *L*_m_ and *L*m_D_ still needs to be confirmed in a larger study.

We acknowledge a number of study limitations, including the retrospective nature of this work and the small sample size: although this sample size would not be sufficient for clinical diagnostics conclusions, it should be sufficient for this methodology development and confirmation study. Generally, acceleration factors are guided by image matrix size, the number of *b*-values, and breath hold duration. For example, the in-plane resolution of multi-b diffusion-weighted ^129^Xe k-space used for the AATD participant (64 × 64) could be matched with multi-b diffusion-weighted ^3^He k-space (128 × 128) used for the same participant by using AF = 2 or half-echo partial Fourier reconstruction. Compressed sensing with at least AF = 3 can be considered to increase the in-plane resolution and number of *b* values. Moreover, acceleration with AF = 2 is not complicated to implement on FGRE acquisitions [53], and image reconstruction is also relatively straightforward. In contrast, the compressed sensing acquisition scheme we utilized cannot be implemented with FGRE methods and likely requires non-Cartesian [54] or pseudo non-Cartesian [55] sampling methods, which are more complex, requiring application of k-space regridding/interpolation and density-weighting algorithms. Recently, an ingenious compressed sensing approach [31] was pioneered that combined undersampling in both spatial and diffusion-sensitising directions using ^3^He MRI with an acceleration factor of 7. It is this type of novel approach that will help translate MRI morphometry methods to clinical use.

In summary, in elderly volunteers and participants with emphysema, we evaluated two different acceleration techniques with diffusion-weighted ^129^Xe MRI. All sampling methods generated ADC and stretched-exponential model lung *L*m_D_ and *L*_m_ values that were strongly related and not significantly different. The results of this retrospective feasibility analysis provide support for the notion that undersampled single-breath diffusion-weighted ^129^Xe MRI may be considered for studies of emphysema in participants.

The clinically-conducted ADC and *L*_m_ measurements should allow for accurate regional probing of the alveolus sizes and surface-to-volume estimates for a wide range of age groups: this information could then be used for the treatment decision and therapy outcome assessments.

## Figures and Tables

**Figure 1 diagnostics-13-01477-f001:**
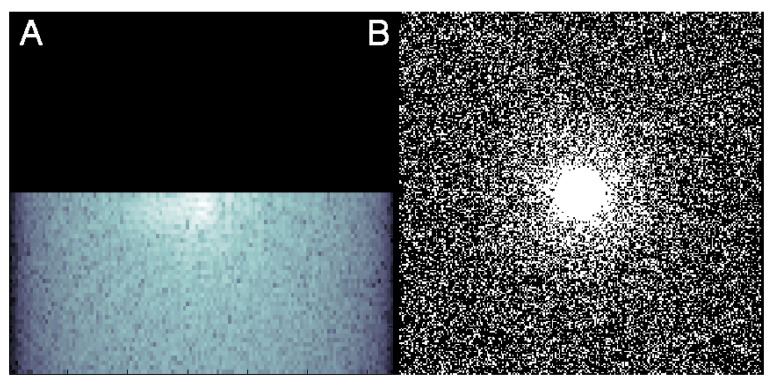
Undersampled k-space masks mimicking accelerated acquisition methods (**A**) Partial Fourier Reconstruction: 50% undersampled k-space in the phase-encoding direction, or AF = 2. (**B**) Compressed Sensing: 66% undersampled k-space, or AF = 3.

**Figure 2 diagnostics-13-01477-f002:**
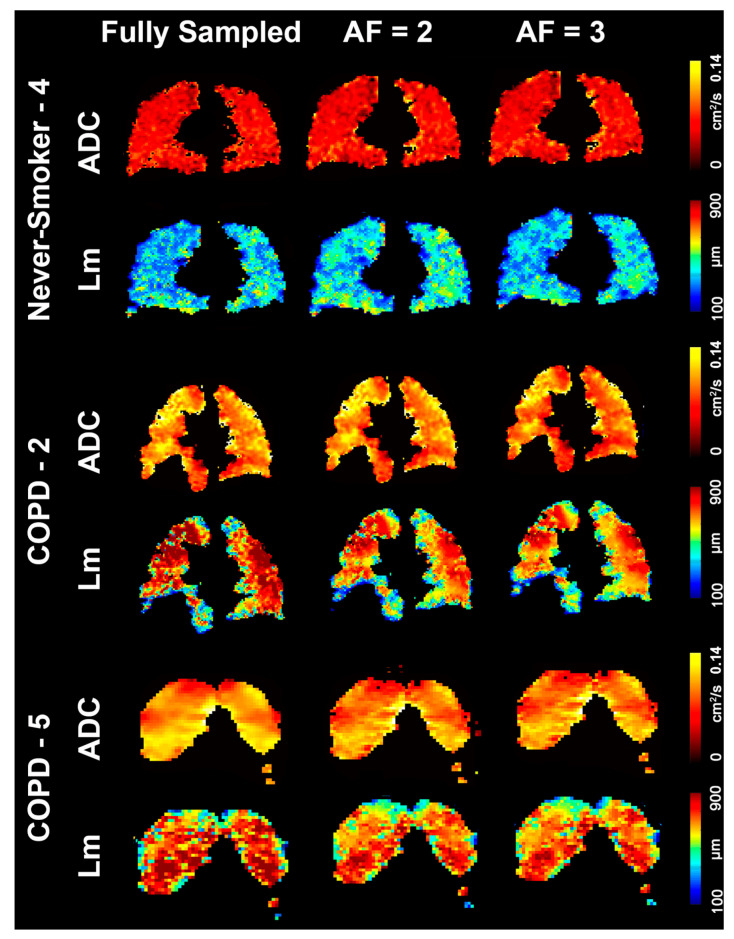
Representative ^129^Xe ADC and *L*_m_ map for elderly never-smoker, COPD participant and AATD participant. Never-smoker-4 is a 71-year-old female with FEV_1_ = 105%_pred_, DL_CO_ = 94%_pred_, RA_950_ = 0.1%, and VDP= 4%. COPD-2 is a 69-year-old male COPD ex-smoker with FEV_1_ = 59%_pred_, DL_CO_ = 43_,_ RA_950_ = 12%, and VDP = 15%. COPD-5 is a 66-year-old male AATD never-smoker with FEV_1_ = 58%_pred_, DL_CO_ = 50%_pred_, RA_950_ = 19%, and VDP = 27%. AF = acceleration factor; ADC = MRI apparent-diffusion-coefficient.

**Figure 3 diagnostics-13-01477-f003:**
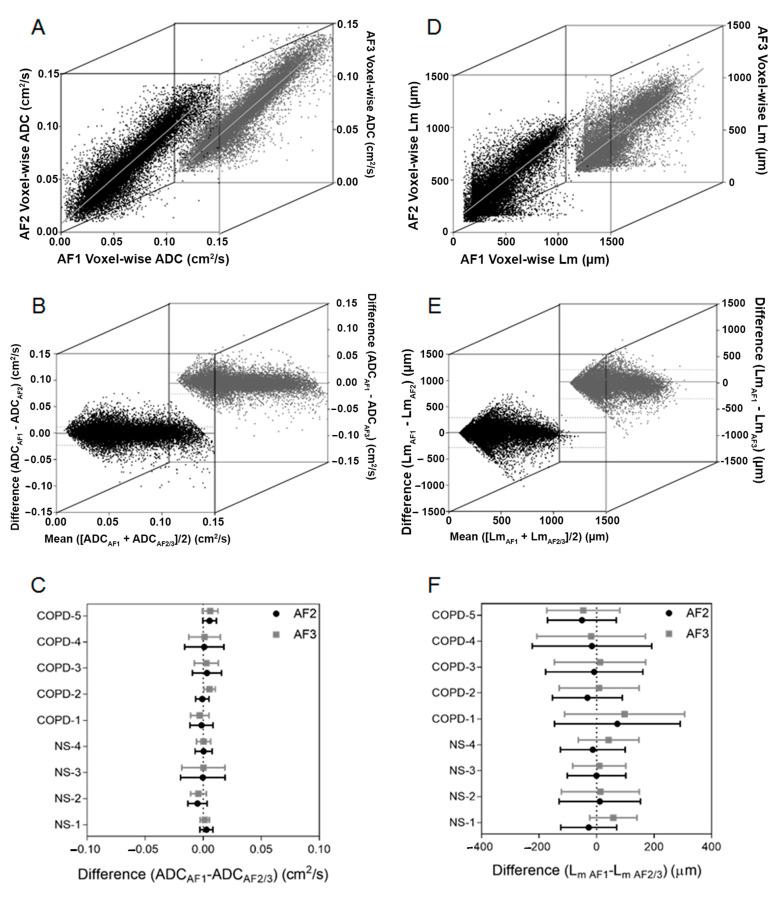
Relationships between ADC and *L*_m_ with acceleration factor. (**A**) ADC; AF-1 was significantly correlated with AF-2 (r^2^ = 0.82, ρ = 0.90, *p* < 0.0001, y = 0.88x + 0.008) black dots, and AF-1 was significantly correlated with ADC A-F3 (r^2^ = 0.86, ρ = 0.92, *p* < 0.0001, y = 0.87x + 0.008) grey dots. (**B**) Bland-Altman analysis of agreement for ADC AF-1 with AF-2 (bias = −0.0002 ± 0.01 cm^2^/s, lower limit = −0.02 cm^2^/s, upper limit = 0.02 cm^2^/s) black dots and Bland-Altman analysis of agreement for ADC AF-1 with AF-3 (bias = −0.0007 ± 0.01 cm^2^/s, lower limit = −0.02 cm^2^/s, upper limit = 0.02 cm^2^/s) grey dots. (**C**) ADC differences for AF-1 with AF-2 (black) and AF-3 (gray) for each participant. (**D**) *L*_m_; AF-1 was significantly correlated with AF-2 (r^2^ = 0.64, ρ = 0.75, *p* < 0.0001, y = 0.79x + 101) black dots. AF-1 was significantly correlated with AF-3 (r^2^ = 0.66, ρ = 0.76, *p* < 0.0001, y = 0.77x + 78) grey dots. (**E**) Bland-Altman analysis of agreement for *L*_m_ AF-1 with AF-2 (bias = 8 ± 146 μm, lower limit = −278 μm, upper limit = 293 μm) in black. Bland-Altman analysis of agreement for *L*_m_ AF-1 with AF-3 (bias = −24 ± 141 μm, lower limit = −300 μm, upper limit = 252 μm) in grey. (**F**) *L*_m_ differences for AF-1 and AF-2 (black) and AF-3 (gray) for each participant. Dotted lines indicate the 95% confidence intervals.

**Table 1 diagnostics-13-01477-t001:** Participant demographic and imaging measurements.

Parameter (Mean ± SD)	Never-Smokers (n = 4)	COPD (n = 5)	Significant Difference *p*
Male Sex n (%)	2 (50)	4 (80)	-
Age years	66 (13)	72 (5)	0.8
FVC%_pred_	102 (8)	100 (21)	0.9
FEV_1_%_pred_	103 (6)	58 (30)	0.1
FEV_1_/FVC%	76 (2)	42 (14)	0.02
RV%_pred_	102 (9)	160 (57)	0.3
TLC%_pred_	102 (8)	123 (8)	0.04
RV/TLC%_pred_	39 (9)	48 (14)	0.9
DL_CO_%_pred_	104 (12)	39 (13)	0.0009
VDP%	4 (0.3)	28 (14)	0.2
RA_950_%	-	19 (9)	-

COPD = ex-smoker with COPD; FEV_1_ = forced-expiratory-volume-1-sec; %pred = percent-predicted; FVC = forced-vital-capacity; RV = residual-volume; TLC = total-lung-capacity; DL_CO_ = diffusing capacity of the lung for carbon monoxide; VDP = ventilation defect percent; RA_950_ = relative area of the CT density histogram ≤−950 Hounsfield units; the significant difference was performed using an unpaired *t*-test with post hoc Holm-Bonferroni correction.

**Table 2 diagnostics-13-01477-t002:** Imaging measurements and morphometry estimates.

	Fully-Sampled	AF = 2	AF = 3	Fully-Sampled—AF = 2	Fully-Sampled—AF = 3	*p*-Value *
**Never-Smokers (n = 4)**						
ADC cm^2^/s	0.05 (0.01)	0.05 (0.01)	0.05 (0.01)	7%	7%	>0.99
*L*m_D_ µm	140 (30)	130 (30)	140 (30)	6%	5%	0.94
*L*_m_ µm	280 (130)	300 (130)	300 (130)	10%	7%	0.59
**COPD (n = 5)**						
ADC cm^2^/s	0.08 (0.02)	0.08 (0.02)	0.08 (0.02)	3%	4%	0.99
*L*m_D_ µm	190 (50)	190 (50)	190 (50)	5%	5%	0.97
*L*_m_ µm	560 (250)	570 (260)	580 (260)	11%	10%	0.95
**AATD (n = 1)**						
ADC cm^2^/s	0.08 (0.01)	0.08 (0.01)	0.08 (0.01)	7%	8%	-
*L*m_D_ µm	220 (40)	230 (30)	230 (30)	6%	5%	-
*L*_m_ µm	690 (210)	730 (180)	730 (180)	13%	12%	-
*p*-value ADC **	<0.0001	<0.0001	<0.0001			
*p*-value *L*m_D_ **	0.003	0.045	0.02	-	-	-
*p*-value *L*_m_ **	0.002	0.009	0.005	-	-	-

* ANOVA between fully-sampled, AF = 2 and AF = 3; ** Independent *t*-test for three different k-space sampling methods (never-smokers vs. COPD). COPD = ex-smoker with COPD; AF = acceleration factor; *L*m_D_ = MRI mean airway length scale estimate; *L*_m_ = MRI mean linear intercept estimate. Note that the AATD participant is also included in the COPD subgroup and values.

## Data Availability

Data are not available due to the ethical restrictions.

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
