# Peer review of "Undersampled Diffusion-Weighted 129Xe MRI Morphometry of Airspace Enlargement: Feasibility in Chronic Obstructive Pulmonary Disease"

_diagnostics, 2023, doi:10.3390/diagnostics13081477_

Round 1

Reviewer 1 Report

The authors evaluated multi-b (0, 12, 20, 30s/cm2) diffusion-weighted 129Xe ADC/ morphometry estimates by Magnetic Resonance Imaging (MRI) using fully-sampled k-space and retrospectively-undersampled k-space with two acceleration-factors (AF=2 and 3) in never-smokers and ex-smokers with chronic-obstructive-pulmonary-disease (COPD) or alpha-one anti-trypsin deficiency (AATD). 

The most interesting result was that multi-b diffusion-weighted 129Xe MRI is feasible using two different acceleration methods to measure pulmonary airspace enlargement using Lm and ADC in COPD subjects and never-smokers. To enhance clinical translation a setting of rapid single-breath acquisitions was successfully used. The results corresponded very well to the lung function measurements in both groups (never-smokers vs. COPD patients) and confirmed the good quality of MRI results.

The study was done in an appropriate matter. All sections (Introduction, Theory, Materials and methods, Results, and Discussion) are written well and contain all informations necessary in a very good informative style of english language. The meanuscript meets all criteria for the publication of scientific studies.

Author Response

Reviewer 1

The authors evaluated multi-b (0, 12, 20, 30s/cm2) diffusion-weighted 129Xe ADC/ morphometry estimates by Magnetic Resonance Imaging (MRI) using fully-sampled k-space and retrospectively-undersampled k-space with two acceleration-factors (AF=2 and 3) in never-smokers and ex-smokers with chronic-obstructive-pulmonary-disease (COPD) or alpha-one anti-trypsin deficiency (AATD). 

The most interesting result was that multi-b diffusion-weighted 129Xe MRI is feasible using two different acceleration methods to measure pulmonary airspace enlargement using Lm and ADC in COPD subjects and never-smokers. To enhance clinical translation a setting of rapid single-breath acquisitions was successfully used. The results corresponded very well to the lung function measurements in both groups (never-smokers vs. COPD patients) and confirmed the good quality of MRI results.

The study was done in an appropriate matter. All sections (Introduction, Theory, Materials and methods, Results, and Discussion) are written well and contain all information necessary in a very good informative style of English language. The manuscript meets all criteria for the publication of scientific studies.

We would like to express our gratitude for the Reviewer’s comments, and appreciate their input and feedback. Some light modifications were still made regarding grammar, and modifications pertaining to the second Reviewer’s comments were also implemented.

Reviewer 2 Report

This study is demonstrated that the multi-b diffusion-weighted hyperpolarized-gas 129Xe MRI measures pulmonary airspace-enlargement using apparent-diffusion-coefficients (ADC) and mean-linear-intercepts (Lm), and is feasible to find the difference of the pulmonary airspace enlargement between COPD subjects and never-smokers using ADC and Lm. This technique is very interesting. However, I have some comments about this study.

Major comments

1. Why were the ADC and Lm by using the multi-b diffusion-weighted hyperpolarized-gas 129Xe MRI different between COPD subjects and never-smokers? Were the ADC and Lm correlated with other lung function test parameters? And, which lung function parameter has the best correlation with the ADC and Lm? 

2. Did the emphysematous areas or volumes affect the data of the ADC and Lm?

3. What are the clinical implications of measuring ADC and Lm for patients with COPD?

Minor comments

1. Is the sample size reasonable? The sample size may be small.

2. Did lung aging affect the data of the ADC and Lm?

3. Is the term of “subject” OK? Authors should consider the use of patient, or participant or individual or with respect to the Research Volunteers.

Author Response

Reviewer 2

This study demonstrated that the multi-b diffusion-weighted hyperpolarized-gas 129Xe MRI measures pulmonary airspace-enlargement using apparent-diffusion-coefficients (ADC) and mean-linear-intercepts (Lm), and is feasible to find the difference of the pulmonary airspace enlargement between COPD subjects and never-smokers using ADC and Lm. This technique is very interesting. However, I have some comments about this study.

Major comments

  1. Why were the ADC and Lm by using the multi-b diffusion-weighted hyperpolarized-gas 129Xe MRI different between COPD subjects and never-smokers? Were the ADC and Lm correlated with other lung function test parameters? And, which lung function parameter has the best correlation with the ADC and Lm?

We thank the reviewer for the comment. Diffusion-weighted Hyperpolarized gas 129Xe MRI permits measurement of the apparent diffusion coefficient (ADC) and alveolus size (Lm), both of which depend on the alveolus geometry.  It is well known that emphysema leads to destruction of the alveolus wall and/or lung microstructure: the gas diffusion is therefore less restricted in the emphysematous lung compared to the normal lung.  The ADC values reflect this microstructure change: they are smaller for normal lungs than for emphysematous lungs. The Lm value is a direct measurement of the alveolus size: the Lm values for normal healthy lungs are between 170µm and 300µm, depending on patient age, while Lm values for severe emphysema patients can be 1000µm.  As gas exchange through the alveolar wall is the main lung function, it is heavily dependent on the alveolus wall surfaces.  The destruction of the alveolus wall directly affects the gas exchange, keeping in mind that Lm is inversely proportional to the surface-volume ratio (S/V) of the alveolus walls: a large Lm value is therefore a result of reduced total alveolus area through which gas exchange can occur.  This suggests a correlation between ADC/Lm and Diffusing Capacity of the Lungs for Carbon Monoxide (DLCO).  Although the correlation between ADC values obtained with 3He MRI and DLCO values has been previously investigated by Kirby et al. (Kirby M, Owrangi A, Svenningsen S, et al On the role of abnormal DLCO in ex-smokers without airflow limitation: symptoms, exercise capacity and hyperpolarised helium-3 MRI. Thorax 2013;68:752-759), we are not aware of any publication reporting a correlation between the Lm and DLCO values: however, we strongly believe that the correlation should be similar to the previously reported one for ADC.

The following passage has been added to the introduction section:

“As emphysema leads to destruction of the lung microstructure and a subsequent decrease in surface area of the alveolar walls, gas motion in the lungs is less restricted leading to higher ADC values.  Moreover, since Lm is inversely proportional to the lung surface-to-volume ratio, destruction or damage of the lung microstructure leads to larger Lm values.  As ADC and Lm values increase, normal/healthy gas exchange diminishes, making these measurements clinically practical for evaluating lung function in emphysema.”

The following have been added to the discussion:

“The correlation between DLCO and ADC values has previously been studied by Kirby et al. [51], but such a correlation has not yet been reported for Lm values.”

With

[51]     Kirby M, Owrangi A, Svenningsen S, et al On the role of abnormal DLCO in ex-smokers without airflow limitation: symptoms, exercise capacity and hyperpolarised helium-3 MRI. Thorax 2013;68:752-759

Along with:

“Several factors are expected to influence ADC and Lm values, such as severity of emphysema and lung aging, due to the fact these measurements are an indirect reflection of the lungs’ and alveoli’s ability to move and transfer gasses: Lm being the lung microstructure dimensions and ADC describing the motion of gasses within the lungs and the airway restrictions. As such, any destruction of the airways and alveoli leads to a change in ADC and Lm values, but the destruction pattern and the distribution of these ADC and Lm values differs from normal lung aging compared to emphysematous lungs [28].”

With

[28]     Paulin, G.A., et al., Noninvasive quantification of alveolar morphometry in elderly never- and ex-smokers. Physiol Rep, 2015. 3(10).

  1. Did the emphysematous areas or volumes affect the data of the ADC and Lm?

We thank the reviewer for this question.  As stated in our previous response, the ADC values reflect the emphysematous change in the lung microstructure, and Lm is a direct measurement of the lung microstructure dimensions.  Therefore, the emphysematous areas are expected to directly affect the ADC and Lm values.  

Please see our previous response for typical values of ADC/Lm for healthy and emphysematous lungs, and for the modifications made to the manuscript concerning this comment.

  1. What are the clinical implications of measuring ADC and Lm for patients with COPD?

The clinically-conducted ADC and Lm measurements should allow for accurate regional probing of the alveolus sizes and surface-to-volume estimates for a wide range of age groups: this information could then be used for the treatment decision and therapy outcome assessments. An example of the clinical implications of measuring ADC is the observation of pure emphysema progression and the assessment of the augmentation therapy efficiency in AADT participants over a 5-year interval, using diffusion-weighted hyperpolarized gas 3He MRI. (Westcott A, Capaldi DPI, Ouriadov A, McCormack DG, Parraga G. Hyperpolarized (3) He MRI ventilatory apparent diffusion coefficient of alpha-1 antitrypsin deficiency. J Magn Reson Imaging. Jan 2019;49(1):311-313.).

Please see the response to the first question for the modifications made to the manuscript addressing this comment.

Minor comments

  1. Is the sample size reasonable? The sample size may be small.

We thank the comment.  We agree with the reviewer that this sample size can be small for a clinical/diagnosis study; however, the sample size used here should be sufficient in this case to verify the method as this paper is focused on method/technique development.

We have modified the following sentence in the discussion section:

“We acknowledge a number of study limitations including the retrospective nature of this work and the small sample size: although this sample size would not be sufficient for clinical diagnostics conclusions, it should be sufficient for this methodology development and confirmation study.”

  1. Did lung aging affect the data of the ADC and Lm?

We thank the reviewer for the question.  The aging effects on lung microstructure of healthy lungs were previously compared to emphysematous lung by Paulin et al. (Paulin GA, Ouriadov A, Lessard E, Sheikh K, McCormack DG, Parraga G. Noninvasive quantification of alveolar morphometry in elderly never- and ex-smokers. Physiol Rep. Oct 2015;3(10)).  This study has shown that healthy aging leads to an increase in ADC/Lm values; however, the lung microstructure destruction pattern is very different from the emphysematous case. While ADC, external radius, internal radius, mean linear intercept, and surface area-to-volume ratio differed in ex-smokers between those with and without emphysema, this was not the case for never-smokers and the ex-smokers without emphysema for whom only the alveolar depth differed.

Please see the response to the first question (major comments 1) for the modifications made to the manuscript concerning this comment.

  1. Is the term of “subject” OK? Authors should consider the use of patient, or participant or individual or with respect to the Research Volunteers.

We thank the reviewer for the suggestion, and have replaced “study subject” with “study participant” across the manuscript.